# Pedagogy of Happiness: A Russian View

**Valerii Chernik \*** , **Iulia Afonkina and Tatiana Kuzmicheva**

Institute of Psychology and Pedagogy, Murmansk Arctic State University, Kapitan Egorova str. 15, Murmansk 183038, Russia; julia3141@rambler.ru (I.A.); tvkuzmicheva@gmail.com (T.K.)

\* Correspondence: chernikval@mail.ru; Tel.: +7-981-301-5912

**Abstract:** For thousands of years the topic of happiness has attracted attention from the representatives of various sciences. However, until now there has been no unity in understanding the essence, sources, and components of happiness. Quite often, the interpretation of the phenomenon of happiness is limited to the analysis of the works of philosophers of antiquity, the European Middle Ages, modern and recent history, and many researchers are in a kind of Eurocentric captivity. Consciously or accidentally, the works of Russian thinkers are often not considered. This hinders the creation of a holistic and more objective picture of such an important aspect of every person's life. In order to overcome this Eurocentric bias, this study suggests tracing back the main stages of the development of the idea of happiness in Russia from the era of Peter the Great to the present. For this purpose, the authors used the methods of theoretical, comparative, and retrospective analysis. The authors believe that currently the pedagogy of happiness is being actively formed. To educate happy people, teachers should master the art of the pedagogy of happiness.

**Keywords:** philosophy of education; pedagogical anthropology; happiness in education; felixology; pedagogy of happiness; free labor

## 1. Introduction

Having received an invitation to take part in a special issue of the journal dedicated to the philosophy of education, we asked ourselves if all authors understand and interpret the essence of this concept in the same way. Independently of others, we will explain what we mean by the philosophy of education.

Our position is largely determined by the ideas of John Dewey, who wrote that "philosophy is the theory of education in its most general aspects" [1] (p. 299). It has a dual task: to revise the past experience and the program of values, and to revive the goals and methods of education in the modern life with its broad and competent review [1] (p. 297).

Sam Harris, a contemporary philosopher, believes that the highest value for people is a maximum of happiness and a minimum of grief [2]. Therefore, all ethical differences can be reduced to the justification of the most effective ways to achieve maximum happiness.

Happiness is not a new research topic. A detailed analysis of the theoretical foundations of understanding the essence of happiness is made in the recent study "The Happy Level: A New Approach to Measure Happiness at Work Using Mixed Methods" by Gisela Sender, Flavio Carvalho, and Gustavo Guedes [3]. Having presented the scientific community with an interesting material for analysis and a better understanding of what happiness is and how to be happy, the authors rightly say that it is important to continue studying a topic that is multifaceted and cannot be limited to one perspective [3] (p. 13).

It seems to us that approaches to the problem of happiness of Russian researchers are interesting. Before describing them, we want to emphasize that the topic of happiness is important for everyone. Surely, everyone understands it really individually, in their own way. An outstanding poet of Russia, Eduard Asadov, wrote a poem about this:

*So, what is the happiness,*
*It's a very good question*
*Somebody says- smoking, cards*
*Alcohol and flirtation.*
*Others say is a beautiful vision*
*Of money and a higher position*
*In flattering kindly permitted*
*And awe of the workers submitted.*
*Third point of view that it is in reflection —*
*Sweet love and respect, good care relation*
*And general hearts' comprehension.*
*To my mind in all enterprise*
*Happiness has different size*
*From peaks to a lowest mound*
*Depending on people around!* [4].

The phenomenon and essence of happiness are not only understood ambiguously, but are also studied in different ways since researchers employ various methods and proceed from a variety of methodological sources. The authors of this article used several research principles, the most important of which is the principle of historicism. Its essence was described by one of the authoritative scientists of Russia, academician Alexey Piskunov (1921–2005). According to him, "the principle of historicism in the pedagogical science, as well as in the science in general, presupposes taking into account the changes of the studied object in the process of its development. The principle of historicism requires analysis of the qualitative features of each stage in the development of the studied pedagogical phenomenon, understanding of their being pre-conditioned by various circumstances, understanding connections between them and their specific mechanisms" [5] (p. 207).

Moreover, the authors were concerned with the idea of Zakhar Ravkin (1918–2004), a corresponding member of the Academy of Education of Russia, who stated that when conducting research, it is important to identify key turning points in the history of pedagogy; to get rid of unnecessary details in the presentation of pedagogical facts and phenomena; and to create favorable conditions for comparative analysis and generalizations [6] (p. 94). Therefore, in this scientific research the concept of "happiness" is described in its evolution in various periods of Russian history, which determined the essence of its understanding.

Another basic principle of this research was developed by Arthur Oncken Lovejoy (1873–1962) and is called "intellectual history", or the "history of ideas". It involves studying the processes of genesis and evolution of human ideas. The author of the concept of "intellectual history" viewed it as "biography of ideas" [7]. The research potential of this approach seems to be quite important and capacious because it is achieved by eliminating disciplinary boundaries and forming interdisciplinary approaches that combine social history, pedagogy, philosophy, religious studies, physiology, psychology, medicine, oral folk art, literary studies, etc. The authors believe that this principle of research enables getting a better understanding of the Russian view on its essence in the description, comparative analysis and interpretation of the genesis of the concept of "happiness" in the history of pedagogy and philosophy of education in Russia. This idea of Arthur Lovejoy in a certain way coincides with the appeal of the founder of scientific pedagogy in Russia, Konstantin Ushinsky (1824–1870), who wrote: "If we want to fully bring up a child, we must first fully understand this child" [8] (p. 23). That is why this study relies on works on history, religious studies, philosophy, pedagogy, psychology, physiology, literary studies, etc.

In recent years, studies have been conducted that try to identify the source of happiness. The results obtained by representatives of the natural science approach are interesting in this regard. According to them, hormones such as serotonin, dopamine, oxytocin, and endorphin make a person happy [9]. It can be assumed that happiness is just the result of the hormones' activity, and a person can cause it independently. It is enough for a person to eat a little chocolate or drink a cup of coffee to provide the body with serotonin and,

consequently, with it a bit of happiness. Do you want to be a little more cheerful? Here are bananas or ice cream, and the endorphin obtained together with them will make you more cheerful. Or, suppose there is a lack of dopamine in the body, as a result of which you experience no anticipation of pleasure and the desire to take on a difficult task. Perhaps it seems to someone that there are no problems here. Others think this problem is easy to fix. Here's a walnut, an apple, some chicken; just eat them and your body will get a dose of dopamine.

According to the results of the research carried out by specialists (L. G. Breuning [10], V. A. Dubynin and K.S. Tanaeva [11], A. G. Rozhkov [12]), the level of hormones of happiness can be increased by using such harmful substances as nicotine, drugs and alcohol. And again we have to answer the question: is this happiness or pleasure? Are they equivalent?

Attempts made to understand the essence, content, and sources of happiness take us back in history again and again. The dichotomy of happiness as "material–spiritual" has already appeared there and then. May this be due to the fact that a person is a unity of the physical and the spiritual? We presume that this kind of dichotomy is one of the reasons for different approaches to understanding the essence of happiness which were developed already in ancient philosophy: hedonism, utilitarianism, and eudemonism.

## 2. Concepts of Happiness in Different Periods of Russian History

### 2.1. Happiness in the Tsarist Russia

The emergence of hedonism is associated with Aristippus (c. 435—356 BC, Greece). His main idea is: enjoy today, enjoy the day and the pleasures that are available, especially the physical ones, do not restrain yourself by any rules [13].

Epicurus (341–270 BC, Greece) treated the goal similarly to Aristippus, i.e., the achievement of happiness. But he understood the means to achieve this goal differently. According to Epicurus, in order to be happy, you need to organize your life, to make it fair and moderate. A person has to consider not only their own needs, but also those of other people. The name of Epicurus is associated with the emergence of utilitarianism [14].

Aristotle (384–322 BC, Greece), as a representative of eudemonism (this is a Greek word commonly translated as 'happiness' or 'welfare'), represented happiness as a whole complex of states, means and actions from which happiness can arise. He wrote "... Everyone thinks that a happy life is a life that gives pleasure, and they quite reasonably include pleasure in [the concept of] happiness, because no active manifestation is complete if it is hindered, and happiness refers to things that have reached fullness. That is why a happy person also needs physical pleasures, external pleasures, and a chance so that not to have any obstacles here" [15].

All of these approaches have been criticized in the Russian science of happiness (felixology) [from lat. felix–happy, bringing happiness].

In Russia already in the Middle Ages, the topic of happiness was not in the center of attention of the authors of the sources that have come down to modern days. This may be grounded in the influence of the Orthodox Church which was strong in Russia. According to its postulates, earthly happiness is illusory and fleeting. In the desire to achieve earthly happiness, a person forgets about the salvation of the soul. This can turn into an eternal misfortune. True happiness lies in the service to and satisfaction of God, in the admiration of holiness and in the hope of eternal salvation.

This is exactly what the leaders of the Orthodox Church thought to be true. For example, John Chrysostom (c. 347–407) wrote: "To determine the happiness of life on the basis of luxury and wealth and material things is peculiar not to those people who pay attention to their own nobility, but to the people who have become horses and donkeys" [16]. In his opinion, "...true happiness... begins only when a person is cleansed of sinful vices by sincere repentance and begins to spend their life as the Holy Orthodox Church teaches" [16].

Up to this day, the fathers of the Orthodox Church inspire people with the idea that "Happiness does not consist in earthly well-being, but in faith in Christ" [17] (p. 693)].

O. E. Kosheleva, a researcher of the problem of happiness, believes that "philosophizing on the topic of happiness is a luxury affordable to those segments of the population

who did not need to dedicate all their life energy to survival, to the struggle for food. For the majority of low-income people of ancient Russian society happiness remained close to the concept of "well-being", it meant achieving economic well-being and getting rid of backbreaking labor" [18].

The socio-pedagogical thought of Russia gives a kind of polar assessments of utilitarianism and eudemonism. Classical utilitarianism, which was formed under the conditions of the English Industrial Revolution, could not achieve the same success in the country of serfdom that stood on completely different ideological, civilizational foundations. Arguments "against" utilitarianism and eudemonism were formulated in the context of the Russian Orthodox Church's ideas and moral theology, adherence to the traditions of the Slavs (Slavophilism). For social and pedagogical thinkers and figures of Russia (V. Odoevsky, V. Solovyov, E. Trubetskoy, etc.), utilitarianism symbolized all negative features of Western civilization. Starting from criticism of utilitarianism, they searched for the foundations of authenticity and identity of the Russian culture. It is curious that opponents invented many synonyms and epithets for utilitarian ethics that have a more or less negative meaning in the Russian language: greed, avarice, philistinism, mundanity, hucksterism, mercantilism, materialism, consumerism. This position is especially clearly expressed in the philosophical heritage of Vladimir Solovyov (1853–1900) [19].

Were these ideas reflected in the pedagogy and education in Russia? Without any doubt, they were. In Russia already in the 18th century, when under Peter the Great (years of reign 1682–1725) the system of public schools began to take shape, the problem of happiness was included in the processes of education and training. Its various aspects (what is happiness, how to achieve happiness, why some people are happy and others are not, etc.) were discussed by child- and adolescent-age students. However, under Peter the Great the education system was focused on technical and special (professional) training. The tsar primarily needed navigators, gunners, ore miners, ship-builders, etc.

Empress Catherine the Great (years of reign 1762–1796) proclaimed a course for the education of a "new breed of people" who would be useful for the state and could change the world for the better. Thus, it can be assumed that by her will and decisive measures Catherine the Great wanted to "make her subjects happy". The queen's own explanation gives us a clue: "Happiness is not as blind as people usually think. Often it is nothing more than the result of correct and firm measures that were not noticed by the crowd, but, nevertheless, prepared a certain event. Even more often it is the result of personal qualities, character and behavior" [18]. At the same time, Catherine the Great believed that the very purpose of education is to achieve happiness for a person.

It is important to remember that the absolute majority of the population of the Russian Empire under the rule of Catherine the Great, who proclaimed the policy of enlightened absolutism, remained illiterate, had no opportunity to get an education. Due to this, serf peasants and ordinary people had their own interpretation of happiness, which is reflected in the folk sayings:

> *Fortune is easily found, but hard to be kept.*
> *Happiness takes no account of time.*
> *Happy is he that is happy in his children.*
> *One is happy that thinks oneself so.*
> *Misfortunes tell us what fortune is.*

By the middle of the 19th century, serfdom still existed in Russia, which allowed the landowner (feudal lord) to sell a serf peasant to another master, to punish at his own discretion, to lose at cards, to use young peasant women as concubines, or use the right of the "wedding night", etc. It was alien to many progressively-minded people, including those from the upper strata of the society. They opposed such rules, and spoke about the need to change the goals and objectives of education.

At that time the pedagogical thought in Russia was awakened by the famous surgeon and teacher Nikolai Pirogov (1810–1881). His article "Voprosy zhizni" ("Questions of Life")

(1856) provoked a desire to think about the goals and essence of education and life itself in the entire Russian society of that time.

In the article, N. Pirogov spoke out sharply against such an interpretation of happiness, which is based on a material, almost commercial desire for pleasure [20]. He mentioned different types of happiness: for some people happiness consists in being able to sleep and eat; for others it is peace of mind, refusal of doubts and reflections; and for others, it is the performance of official duties for their own benefit, saving money for a rainy day and abandoning beliefs. At the same time, N. Pirogov was ironic saying, "With a full pocket, you can live without beliefs" [20]. There is the happiness of a conformist, there is the happiness of a "worm on a pile of dirt", etc. [11].

According to N. Pirogov, happiness is an art, and it can be achieved by labor and talent. The task of the education is to develop these abilities. Only this can "awaken a human in a person" [20]. To do this, it is necessary to develop an inner person, i.e., a person with beliefs, which can be facilitated by a humanitarian, universal education. This education is able to realize the task, that is "to first learn to be a human" [20]. "...nowhere is there so much education for the inner person as in learning, not for nothing called humane", the great surgeon and citizen of Russia believed [20]. To abandon universal education in favor of early specialization means "to open the door for rude charlatanism to an ill-mannered society and to give shelter to speculation on ignorance and credulity" [21] (p. 133). Therefore, it is important to teach a person from their early childhood to look into themselves shrewdly, to love the truth sincerely, to stand up for it firmly and to be naturally frank with both mentors and peers [20].

Konstantin Ushinsky (1824–1870), who is considered in Russia to be the founder of Russian scientific pedagogy, also paid attention to the education and upbringing, first of all, of a person in a person. To solve the problems of education successfully, you need to study a person from different perspectives. Konstantin Ushinsky himself laid the foundation for a comprehensive science of human education based on its comprehensive study of it–on pedagogical anthropology. His main pedagogical research work is called "Chelovek kak predmet vospitaniya. Opyt pedagogicheskoj antropologii" ("Person as an object of education. The experience of pedagogical anthropology"). In it, he wrote that a person has a desire for happiness from birth. And this is not a desire for pleasure at all, since a person "by an innate desire for happiness can strive to satisfy such aspirations, the satisfaction of which does not give them pleasure at all" [22] (p. 348).

K. Ushinsky developed this idea in his article "Trud v ego psihicheskom i vospitatel'nom znachenii" ("Labor in its mental and educational value"). Here, he writes that pleasure is not happiness at all. Adam and Eve may have been happy while they were in heaven and did not commit their first sin. Having expelled the first people from heaven for their fall, the Lord gave them labor, and there is no happiness for a person without labor. "Labor is an only thing available to a person on earth and the only happiness worthy of people" [23]. It can be concluded from this that a person should not be brought up for happiness, but a person should be prepared for life-long free-will labor.

It is very important to understand that not every activity can be called labor. According to K. Ushinsky, labor is "such a free and consistent with Christian morality activity, to which a person agrees because of the unconditional need for it to achieve one or another truly human goal in life" [23]. Therefore, defining the main goal of education as the happiness of the student, K. Ushinsky writes that at the same time that the educator should in no case mix up happiness with pleasure, but see in happiness "a free, endless and progressive activity corresponding to the true needs of a human soul, and in pleasures–only side effects that may accompany this activity, and may not accompany it" [22] (p. 357). At the same time, for practicing teachers (kindergarten teachers, school teachers, university teachers), K. Ushinsky's idea that every person must be involved in both physical and mental labor is important.

The ideas of N. Pirogov and K. Ushinsky became decisive for the development of pedagogical theory and educational practice in Russia, even in the period after the Great

October Socialist Revolution of 1917, when the Communist Party led by Vladimir Lenin came to power. It was believed that the power of the Bolsheviks (the Communist Party) reflected the interests of working people, ordinary workers, and proletarians of both physical and mental labor. Naturally, in these conditions, happiness was closely associated with creative labor of a person.

### 2.2. Happiness in the Soviet Russia

In the Soviet Russia, the ideas of Anton Makarenko (1888–1939) were popular and significant at the same time. Efficiency and progressiveness of his ideas can also be confirmed by the fact that by the decision of UNESCO (1988), the Soviet teacher A. Makarenko was listed among four teachers (alongside with an American John Dewey (1859–1952), a German George Kerschensteiner (1854–1932) and an Italian Maria Montessori (1870–1952)), who determined the way of pedagogical thinking in the 20th century.

By the beginning of the 1920s, when Anton Makarenko began his pedagogical activity, Russia faced the most acute problem of that difficult time. The Civil war, famine, devastation and banditry that swept the country gave rise to a huge number of street children, deprived of parents, their home, and life values. Child homelessness took the scale of a gigantic disaster.

Many progressively-minded people in Russia understood that children needed protection. "Return childhood to children!", "Everyone to help children!"– the best representatives of Russian intelligentsia responded to these slogans. At the end of 1918, the first public organization appeared in Poltava, the League for the Salvation of Children. It was led by the famous writer and publicist Viktor Korolenko (1853–1921, Russia and Ukraine). Members of the "League" raised funds to help the homeless, organized kindergartens, foster homes, colonies, and sanatoriums in which children were fed and medically treated. Thanks to these efforts, thousands of under-privileged children were saved.

To overcome this social disaster, numerous orphanages, colonies, and communes started to be created. In September 1920, Anton Makarenko was offered to lead the work on the organization of a colony for young offenders.

From the very beginning of his work with these teenagers, Anton Makarenko, referred to as the "father of street children", applied the principle of "burned biography": complete disregard for the past and even more so for the past crimes. At the same time, he himself was well aware of the former life of the colonists, even when he claimed that he asked his superiors not to send him more "cases". On the basis of "cases" and personal observations, A. Makarenko made notes about the colonists, and these notes remained taboo not only for colonists, but even for his colleagues.

A. Makarenko understood that an only way to normalize the activities of the colony, as well as to improve the living conditions, was joint labor. It was very difficult to imagine how to train for creative labor those people who were used to a slutty and criminal lifestyle.

But Anton Makarenko managed to motivate the adolescents. A year later, the colony and the life of the colonists changed for the better.

"A happy childhood does not mean a carefree one," A. Makarenko believed, and therefore, –"the foundation of the Russian school should not be grounded in labor-work, but labor-care" [24]. In this contrast, one can trace back an idea of the above-mentioned K. Ushinsky, who in his article "Labor in its mental and educational meaning" characterized labor-work as something that lowers a person to the level of an animal or a slave when this work is not free, does not have a creative beginning, does not bring pleasure, and does not develop the person. We need free labor, to which a person agrees for the sake of achieving a great goal and which, alongside with bringing pleasure, contributes to the development of an individual.

Anton Makarenko put the idea of "labor-care" into practice. Thanks to this, not only the financial situation of the colony changed: they grew bread and vegetables; they obtained and began to breed thoroughbred cows and pigs; they set up the operation of the mill, etc., but also the main changes occurred in the adolescents, who began to recognize themselves

as fully-fledged owners of their production, which they were learning to manage. When working together, young people formed special relationships linking them by a common responsibility and a common desire to achieve their goals [24].

Anton Makarenko needed 16 years of work, 16 years of need and overcoming to organize such production (we want to mention that this is a lot but still less than the 40 years during which Moses led his people through the desert in search of a better life for them). According to A. Makarenko, "any children's collective, if it wanted to switch to serious production, would also spend at least 10 years, and, of course, the first generations who would fight for this production would quit without having experienced all the benefits yet. The next ones would try them. We need not think that the first generations would quit being offended. After all, working for the sake of a good goal set for future years is already worth a lot in terms of qualifications and education. Perhaps, in this whole process, the main thing is this collective activity, this striving forward, the march towards clearly set goals" [24].

Shortly before his unexpected death from a cardiac rupture (he was 51 years old), Anton Makarenko shared his plans with the audience of a public lecture: he dreamed of writing a book "on such a topic: how to educate a person so that they, whether you like it or not, became happy". There came a question from the audience: "We really want to be happy. Please tell us what you are going to write on this issue". Anton Makarenko replied: "... I expect that the first rule of the wisdom (how to be happy) is: do not rush into happiness, as you are in a hurry... You need to approach this very carefully..." [24].

Of course, many components of happiness are laid in the family. According to Anton Makarenko, "every parent wants their child to be happy. This is the goal of parent' life. For this purpose, parents are ready to give up their own happiness, they are ready to sacrifice their own happiness, if only their son or daughter were happy, too. It is very difficult to find such parents who would not think about it and would not want it" [24]. However, a very difficult question arises: what traits of character, on what habits, traditions, development, beliefs does happiness depend on and what is happiness?

First of all, the parents themselves should be happy. Only a happy person can raise their child being happy, too. However, sometimes parents are ready to sacrifice their own happiness—"for the sake of the child's happiness". When doing so they believe "...I, the mother and I, the father, give everything to the child, sacrifice everything to the child, including our own happiness" [24]. According to Makarenko, this is unacceptable. He wrote: "The most terrible gift that parents can give to their child... if you want to poison your child, give them a large dose of your own happiness to drink, and he will be poisoned" [24].

A. Makarenko insisted that parents should not sacrifice themselves. Humiliation should not be allowed. Children should certainly think about their parents' happiness, and children's desire to bring happiness to their parents should be cultivated. "In the children's eyes the father and mother should have the right to happiness in the first place. There is no sense either for mothers or for daughters, and even more so for the state to educate consumers of maternal happiness. The most terrible thing is to raise children at the expense of maternal or paternal happiness", Anton Makarenko wrote [24].

More than 80 years have passed since the death of A. Makarenko, but the influence of his pedagogical ideas remains strong in Russia. Anyway, we have to admit the fact that in different periods of the history of Russia and the Soviet Union as a whole, the attitude towards A. Makarenko and his ideas was like a pendulum: he was elevated, then relegated; he was elevated to a kind of Olympus of Russian teachers, then some people tried to present him as a miserable pervert. The authors of this article, having devoted a lot of time to studying the heritage and experience of Anton Makarenko, are convinced that Makarenko himself devoted his life to the happiness of people.

Folk wisdom says that the meaning and happiness of a human life is in a planted tree, a built house, a raised son. The practical result of A. Makarenko's activity was that he helped more than three thousand former criminals and street children to find sense in

their lives. He devoted his whole life, mind, and talent to these children and teenagers; he taught them to live as real people should. All graduates of the colony and the commune became honest and morally decent people and excellent workers. It is not accidental that many countries show an increasing interest in the theoretical heritage and practical experience of Anton Makarenko. In the modern conditions of increasing segregation specialists and simply concerned people are interested in how Makarenko managed to return homeless, sometimes feral children to society but also to build a viable children's collective, achieving irreversible positive consolidation of their life. Many people are attracted by Anton Makarenko's belief that a normal person cannot adapt to a "social garbage dump", in the conditions of a garbage dump person grows into a fittest bastard.

*2.3. Happiness in the Post-Soviet Russia*

As a response to global social changes, the new millennium brought new trends to education, which significantly changed the landscape of pedagogical science and determined the directions of the necessary scientific understanding of various social phenomena from the perspective of a developing and emerging personality. The pedagogy of happiness can be considered one of these trends.

In our opinion, research in this field will make it possible to get a more holistic view of the values, goals and objectives of education in a transitive society, as well as to expand the philosophical picture of the modern world.

A Russian researcher A. Subetto [25] understands that the concept of "pedagogy of happiness" arises, in a broad sense from the creative ontology of the world and creative philosophy, and in a narrow sense from the culture of joy understood as a culture of creativity and health. The author offers an interpretation of the pedagogy of happiness through the behavior of a teacher—a smile, laughter, humor—which demonstrates the teacher's ability to be happy in the current situation of educational interaction with students. Focusing on specific forms and methods of such interaction, the author draws attention to their cooperation and co-creativity. These processes do not only help to transfer and acquire knowledge, but also form students' completely different views of the surrounding world, based on the perception of the miracle of the secrets of nature and man.

Thus, the pedagogy of happiness should be understood as a component of the pedagogical science and an element of the philosophical view of the world of a modern person. Such an approach expands the interpretation of the role of pedagogy and education in building a view of the world as a theoretical model of being correlated with the spiritual and social activity of a person, revealing the ways of cognition of reality.

Based on the above-mentioned information, it is obvious that the pedagogy of happiness is supposed to increase the number of people for whom joint actions, relationships, and joint creativity will be a source of joy associated with a positive attitude to the world and themselves and to encourage the development of effective life strategies that reflect a positive image of the future. This creates, in a certain sense, an alternative to hedonistic culture as a culture of pleasure. In its turn, a positive image of the future determines good social well-being in the present.

Thus, happiness as an active human emotion provides a connection between the present and the future on the basis of the unfolding and enriching creative potential of an educated and developing person.

We emphasize that an unhappy teacher will not make students happy. However, not every happy teacher will make them happy either. It is not enough for a teacher to demonstrate joy by themselves. The teacher should experience the involvement in the life of their students, concern with their difficulties and problems, empathic response to the "movements of their soul" in order to teach children to be happy.

Today, the pedagogy of happiness in Russia is in search of those real "bonds" that will bring together and unite the world of children and the world of adults, without opposing them and trying to find out who has more rights and authority: a wise teacher or a child entering life.

A student cannot be taught happiness similarly to a writing or reading skill, but a student can be taught to notice and experience happiness in joint activities and creativity. In this sense, happiness means freedom of self-expression in a social community where a child is accepted as they are. And children, in their turn, seek to learn, make discoveries, understand themselves and others when they create an image of the desired future.

Thus, E. Yanakieva considers that [26], for students, favorable interpersonal relationships become the main thing. A happy child strives to achieve success and realize themselves, has a positive self-esteem, infects others with their positive feelings and creates an environment that supports its development, guaranteeing a feeling of harmony and satisfaction.

For the teacher, children's happiness acquires an axiological role in determining all the components of education, unveiling the inner world of a child to him and acting as the highest value.

The pedagogy of happiness involves understanding the role of creativity in the education of children. In our opinion, the most important point connected with children's happiness is that creativity gives freedom of choice and freedom of action for self-expression. Such freedom is a procedural side of the activity of a teacher who wants to teach a child to be happy. We emphasize that the creativity of a student rarely appears by itself as it requires the use of certain pedagogical techniques. Looking back from the modern perspective at the heritage of outstanding Russian teachers, we want to mention different related traditions.

A well-known and respected teacher in Russia Vasily Sukhomlinsky (1918–1970) stated: "The alpha and omega of my pedagogical science is a deep belief that a person is what their idea of happiness is" [27]. Reflecting on what spoils a child's personality, why a child becomes lazy and naughty, Vasily Sukhomlinsky believed that this is because the child does not know the happiness of labor. If children are taught to work, to value labor, then they will value their privilege, will love labor. "To give children the joy of labor, the joy of success in learning, to awaken in their hearts a sense of pride, self-esteem–this is the first regulation of education" [28]. Answering the question about the essence of happiness, Vasily Sukhomlinsky wrote that it is, first of all, optimism, faith in the bright future, creative work, experiencing the joy of satisfying the most vital need–the need for a person. By that he meant the luxury of human communication also mentioned by Antoine de Saint-Exupery in his famous "The Little Prince".

According to Vasily Sukhomlinsky, the sources from which the younger person drew his joys in the childhood determine the moral character of the individual. Having set a goal to create a "school of joy" and having successfully implemented it, he applied such means as games, languages, fairy tales, metaphors, music, painting, nature, and labor as components of the child's spiritual world. The scale of these pedagogical tools, their kind of universality makes their use by the teacher important for the modern pedagogy of happiness.

Any creativity, no matter how we look at it through the prism of individual self-expression, is collective in its essence. Therefore, analysis of the creativity of Russian teachers of the past, in particular, Vasily Sukhomlinsky, allows us to determine the determinant of "needs for others". In creative activity, the child is required to have the will and to overcome obstacles. This is not a strenuous challenge but the "work of the soul", which is a source of broad positive emotions, when the happiness of the present reinforces the happiness of the future. According to V. Sukhomlinsky's pedagogical concept, the desire to do good to people, to see and create beauty, the joy of communication, knowledge and work–these, according to his pedagogical concept, are the grounds for raising a happy person.

Answering the question, what was the most important thing in his life, Vasily Sukhomlinsky wrote: "I answer without hesitation: love for children" [28], which, in his opinion, is inseparable from the pedagogical culture. This quote can be regarded as a commandment for the pedagogy of happiness.

## 3. Conclusions

The analysis of the development of views on the problem of happiness at different stages of the historical development of Russia allows us to draw some conclusions.

First: the topic of happiness has attracted attention of thinkers, researchers, heads of state, and religious figures for many centuries. The topic of happiness and a person's preparation for it was institutionalized in the field of education in the era of Peter the Great with the emergence of a system of public schools. However, the understanding of happiness was polarized: the rich, the upper classes did not understand it in the same way as ordinary, poor people.

Second: in the middle of the 19 century, under the influence of the socio-pedagogical movement, by the initiative of progressive thinkers of Russia, the topic of human happiness acquired a new meaning, was linked to the understanding of the need for free, creative labor that contributes to the development of the personality of each individual, regardless of their origin and social status. The criticism of attempts to identify or associate happiness with pleasure, idleness was increasing. The idea of the importance of education for a person's understanding of genuine happiness was reinforced.

Third: in the Soviet period of Russia's history, largely thanks to the efforts of Anton Makarenko, the theoretical foundations of the pedagogy of happiness were developed, and it was tested in special educational organizations, including penitentiaries. Its use helped thousands of people who had become offenders, criminals due to various social conditions of life to re-socialize, to obtain a truly happy and decent life.

Fourth: currently, a new direction of the philosophy of education is developing in Russia called felixology (a section of the philosophy of education that studies various aspects of the theory of happiness). On its basis, various aspects of the pedagogy of happiness are pointed out. We are absolutely convinced that only a happy teacher can raise happy students. To do this, it is necessary to familiarize with a new generation of teachers to the fundamental ideas related to the essence of happiness as one of the most important values of human life.

Fifth: a retrospective analysis of the idea of happiness as a value, the experience of its implementation in different epochs does not only allow us to better understand the essence of the problem, but also to revive the goals and methods of education in the modern life.

We believe that the article will help international researchers to get an idea of how happiness has been understood and interpreted in different historical periods in Russia. Despite the uniqueness of the history of Russia, the specifics of its national mentality, the reader can feel that the topic of happiness in Russia has attracted the attention of a variety of thinkers. We have focused on the views of those who have particularly influenced the development of the pedagogical theory and practice of education in Russia. We hope that this is the first step towards overcoming a kind of refusal to take into account the position of Russian scientists in the study and understanding of the specifics of the phenomenon of happiness.

How can we overcome the notorious Eurocentrism in this case? To our mind, positive experience of cooperation between Norwegian and Russian scientists can be used as an example. The need for a better understanding of the general problems of improving life in the Arctic has prompted teachers (and researchers) of The Arctic University of Norway (Alta campus) and Murmansk Arctic State University to bring to life a multidimensional dialogue. During frank, mutually interested and respectful to the opinion of an interlocutor (not even an opponent!) dialogues the ideas of joint projects with the participation of students were born. An article was also written [29] as an example of a dialogue in science regardless of the researchers' country of residence or affiliation. In this sense, science has no citizenship–it is international. This fully applies to the philosophy of education. We are sure that the collection of works by various authors will give an impulse to the development of new ideas for the sake of improving education and a happy life for new generations.

**Author Contributions:** Conceptualization, V.C., I.A., and T.K.; methodology, V.C., I.A., and T.K.; project administration, V.C., T.K.; supervision, V.C., T.K.; writing–preparation of the original draft, V.C., I.A.; letter-viewing and editing, V.C.; project administration, T.K. All authors read the prepared version of the manuscript and agreed with it.

**Funding:** This research received no external funding.

**Institutional Review Board Statement:** Not applicable.

**Informed Consent Statement:** Not applicable.

**Conflicts of Interest:** The authors declared no potential conflicts of interest with respect to the research, authorship, and/or publication of this article.

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
