# Peer review of "Pedagogy of Happiness: A Russian View"

_education, doi:10.3390/educsci11090503_

Round 1

Reviewer 1 Report

Happiness is indeed accepted as a utilitarian virtue, but utilitarianism its self is at stake here. No reference at all to any discussion about utilitarianism? What about other virtues?

Author Response

Dear Sir/Madam,

First of all, let us express our gratitude to you for reviewing our article, as well as your opinions and recommendations.

You wrote: Extensive English language and style editing required

We have done linguistic and stylistic editing of the text.

You wrote: Happiness is indeed considered a utilitarian virtue, but utilitarianism itself is at stake here. No mention of utilitarianism? What about other virtues?

The fate of utilitarianism in Russia was not easy; the popularity of this theory has experienced both dizzying ups and downs.

Articles and fragments of books by I. Bentham were first published in Russia in 1804.

Later, in 1866, the book Utilitarianism by J.St. Mill was published in Russian.

The majority of Russian society (philosophers, religious leaders, writers) began to criticize the ideas of utilitarianism. We tried to describe this in our article. But the main attention was paid to the description of the development of views on the concept of "pedagogy of happiness". This was the subject of our research. In the pedagogical scientific works of  N. Pirogov, K. Ushinsky, A. Makarenko and others who are mentioned in the article, virtues are presented that are opposed to utilitarianism proper, eudemonism, hedonism.

In addition, in accordance with the recommendations of other experts, we have added material on the views of religious leaders of the Orthodox Church. Their ideas have had (and continue to have) an impact on the practice of upbringing and education in general.

We thank you again for reviewing. Your comments forced us to come back to the article, to think about and comprehend our views.

Best regards,

Chernik Valerii.

Reviewer 2 Report

This is a well-written paper that clearly presents the development of learning theory associated with the topic of happiness across the centuries in Russia. While a good read for those with an interest in learning science, it could also be enjoyed by the non-technical reader. A worthy contribution to our literature.

I would only suggest that it be clearer regarding the methods used and how they are applied in the manuscript. They are mentioned only briefly.

Author Response

Dear Sir/Madam,

First of all, let us express our gratitude to you for reviewing our article, as well as your opinions and recommendations.

You wrote: English language and style are fine/minor spell check required

We have done linguistic and stylistic editing of the text.

You wrote: I would only suggest that it be clearer regarding the methods used and how they are applied in the manuscript. They are mentioned only briefly.

Following your recommendation, we have added a description of the main methodological ideas that have guided our research. These are the ideas of authoritative scientists of Russia (Academician A. Piskunov, Corresponding Member of the Academy of Pedagogical Sciences of Russia Z. Ravkin). In addition to them, it is also A. Lovejoy and his concept of "history of ideas".

We would like to thank you again for your reading of our research and for your Comments and Suggestions for Authors.

Best regards,

Chernik Valerii.

Reviewer 3 Report

While the overall merit is average (see above), the assessment criteria above do not take into consideration if the submission is appropriate. In my opinion, this submission's content is not relevant to this journal. Instead of focussing on changes to the way school education can be delivered, nearly the whole submission is offering the rationale or explanation for why educational research is important.

So, I guess that a submission to another more appropriate journal would be useful. 

As the authors go about revising the submission, the following few points may be useful:

p2 line 66: This looks strange: "No problem! (?)"

p2 line 69: Who are these specialists in the sentence "according to the results of research by specialists". Shouldn't they be cited?

p3 lines 111-112: Needs a citation or two here.

p.4 line 164: "It is it that is able to realize" does not look right.

p9 line 466: "will give an impulse to" is not understood ...

Author Response

Dear Sir/Madam,

First of all, let us express our gratitude to you for reviewing our article, as well as your opinions and recommendations.

You wrote: Moderate English changes required

We have done linguistic and stylistic editing of the text.

You wrote: Although the overall score is average (see above), the above scoring criteria do not take into account whether an application is appropriate. In my opinion, the content of this message has nothing to do with this magazine. Rather than focusing on changes in the way school is delivered, almost the entire article offers a rationale or explanation for why educational research is important.

We take the liberty of disagreeing with the opinion of the reviewer.

First, in our opinion, the philosophy of education presupposes the comprehension of the problems of both pedagogical theory and educational practices. Even Johann Herbart considered one of the most important tasks of pedagogical science is - to develop pedagogical thinking, but not to train on the ability to reproduce other people's technological methods, no matter how good they are in the experience of others. In this regard, it is important for us to note the remark of the psychologist and specialist in the field of pedagogical education M. Rubinstein that it is impossible to reduce the training of a teacher only to the study of methodological techniques. This is the same as reducing the training of a doctor to training "paramedic".

Secondly, we believe that the study of problems of education and pedagogical theory outside the historical context is erroneous and, moreover, dangerous, because it leads to primitivism and limited thinking of the teacher, however, not only of the teacher.

Even the ancients, for example, Confucius, believed: In order to learn the new, it is necessary to study the old.

The great minds of the 20th century also spoke about the importance of studying history. For example, Winston Churchill said: If we quarrel between the past and the present, we will lose the future.

Hermann Hesse: To live in spirit only in the present is intolerable and senseless; spiritual life in general is made possible only through a constant connection with the past, with history.

This series of statements can be continued.

You wrote: p2 line 66: This looks strange: "No problem! (?)"

We have changed the text based on your opinion.

You wrote: p2 line 69: Who are these specialists in the sentence “based on the results of the research of specialists”. Can't they be quoted?

We are very sorry that the reviewer may not have paid attention to the references after the manuscript. We have mentioned few names of researchers. Taking into account the recommendations of the reviewer, we have added the names of the scientists:

  1. Breuning. Habits of a Happy Brain Retrain Your Brain to Boost Your Serotonin, Dopamine, Oxytocin, and Endorphin Levels;
  2. Dubynin and K. Tanaeva. Maternal depression: when happiness is not a joy;
  3. Rozhkov. About hormones of happiness..

You wrote: p3 lines 111-112: Need a quote or two here.

We agree with this recommendation. We have added several quotes from the texts of Fathers of the Orthodox Church and theologians.

You wrote: p.4 line 164: "It is it that is able to realize" does not look right.

We have transformed this phrase – «This education is able to realize the task – "to first learn to be a human" [11].

You wrote: p9 line 466: "will give an impulse" did not understand ...

In turn, we cannot understand this comment of the reviewer.

We are confident that the collection of works by different authors will give impulse (impetus) to the development of new ideas to improve education and a happy life for new generations.

In our deep conviction, getting acquainted with results of scientific research of any scientist, a person expands their understanding of a problem, they may have their own thoughts about what they has read - it may be agreement or, conversely, an objection. Is it possible that the collections of articles published in a scientific publication do not make a person want to discuss with the author? We believe that this gives a certain impetus to new scientific research. It is in this context that we wrote that articles in the special issue of the journal "Pedagogical Sciences" can give impetus to new scientific research.

We thank you again for reviewing. Your comments forced us to come back to the article, to think about and comprehend our views.

Best regards,

Chernik Valerii.

Round 2

Reviewer 1 Report

I am satisfied with the corrections!

Reviewer 3 Report

Thank you to the team of authors for your careful considerations of comments made, and for making appropriate and suitable changes in the manuscript. Although my second comment did not imply that education researchers should not pay attention to historical perspectives and experiences, I have liked what the authors argued about historical perspectives. I can't agree more as far as this aspect of educational research is concerned.